# Improving the Representation of Whitecap Fraction and Sea Salt Aerosol Emissions in the ECMWF IFS-AER

**Samuel Rémy [1,\*] and Magdalena D. Anguelova [2]**

1  HYGEOS, 59000 Lille, France
2  Naval Research Laboratory, Remote Sensing Division, Washington, DC 20375, USA; magdalena.anguelova@nrl.navy.mil
*  Correspondence: sr@hygeos.com

**Abstract:** The European Centre for Medium-Range Weather Forecasts (ECMWF) operates the Integrated Forecasting System aerosol module (IFS-AER) to provide daily global analysis and forecast of aerosols for the Copernicus Atmosphere Monitoring Service (CAMS). New estimates of sea salt aerosol emissions have been implemented in the IFS-AER using a new parameterization of whitecap fraction as a function of wind speed and sea surface temperature. The effect of whitecap fraction simulated by old and new parameterizations has been evaluated by comparing the IFS-AER new sea salt aerosol characteristics to those of aerosol retrievals. The new parameterization brought a significant improvement as compared to the two parameterizations of sea salt aerosol emissions previously implemented in the IFS-AER. Likewise, the simulated sea salt aerosol optical depth and surface concentration are significantly improved, as compared against ground and remote sensing products.

**Keywords:** aerosol; sea spray aerosol; remote sensing; whitecap fraction; CAMS; IFS; air–sea interaction; surface fluxes

## 1. Introduction

The Copernicus Atmosphere Monitoring Service (CAMS; see Abbreviations for a list of acronyms), operated by the European Centre for Medium-Range Weather Forecasts (ECMWF) on behalf of the European Commission, operationally provides twice daily near-real-time global analyses and 5-day forecasts of aerosols, trace gases, and greenhouse gases since 2014. In September 2018, CAMS also released reanalysis of atmospheric composition [1], which has been continually updated since then and now covers 2003 to 2020. These global analyses and forecasts are provided by ECMWF's Integrated Forecasting System (IFS), which combines state-of-the-art meteorological and atmospheric composition modelling together with the data assimilation of satellite products.

The IFS originated as a numerical weather prediction system dedicated to operational meteorological forecasts. The IFS with atmospheric composition extensions was first developed in the framework of the Global and regional Earth-system Monitoring using Satellite and in situ data project (GEMS; 2005 to 2009; [2]). This development continued with the Monitoring Atmospheric Composition and Climate series of projects (MACC, MACC-II, and MACC-III; 2010 to 2014). CAMS is the current (2014 to present) framework for further developments. Regular model updates and implementations of new parameterization schemes are released as IFS cycles (e.g., cycle 47R1 or CY47R1, where 47 is the consecutive number of system updates and R1 stands for revision number 1). With the extensions implemented so far, IFS is capable to forecast and assimilate aerosols [3–5], greenhouse gases [6,7], tropospheric reactive trace gases [8–10], and stratospheric reactive gases [10]. "IFS-AER" denotes the IFS extension with an aerosol scheme used to provide global aerosol products in the CAMS project.

Here, we focus on the modeling of the sea salt emissions in the IFS-AER. Sea salt emissions are obtained from sea spray source function (SSSF), which gives the produc-

tion rate of sea spray droplets per unit area of sea surface per increment of particle size ($m^{-2}\ s^{-1}\ \mu m^{-1}$), usually at relative humidity (RH; see Abbreviations for a list of abbreviations) of 80%, $dF/dr_{80}$ (see Abbreviations for a list of variables). Two functions comprise the size-dependent SSSF [11]:

$$\frac{dF}{dr_{80}} = f(p_1, p_2, \ldots)g(r_{80}) \tag{1}$$

where $g(r_{80})$ is a shape function (or shape factor) representing the sea spray size distribution and $f(p_1, p_2, \ldots)$ is a dimensionless scaling factor (or magnitude) containing possible dependencies of the sea salt aerosol production flux on environmental forcing parameters $p_1, p_2, \ldots$, including wind speed. There are different methods to develop the specific functions in (1). The most widely used is the whitecap method, in which the whitecap fraction $W$ (a number between 0 and 1) serves as a scaling factor.

The goals of the study are to (1) incorporate in the IFS-AER module new whitecap fraction parameterization based on a year-long database of satellite retrievals, (2) evaluate the effect of different SSSF implementations on the IFS-AER performance for global aerosol predictions, (3) use the results to gain insights on further improvements of the IFS-AER module and the whitecap fraction retrievals, and (4) investigate how different representations of the scaling and shape factors in (1) affect the IFS-AER predictions of sea salt emissions and, with this, the global aerosol products. The implication of this work is to improve the modeling of the air–sea fluxes, with potential benefits for representing the boundary conditions of coupled ocean–atmosphere models such as IFS.

In the following, we start with a description of the main characteristics of the IFS-AER (Section 2.1) in order to show the new elements; we then describe the sea salt emission schemes used in the IFS-AER (Section 2.2). We use in situ observations and satellite retrievals to evaluate the simulations of whitecap fraction and sea salt characteristics (Section 2.3). The results of the simulation experiments (Section 3.1) are tabulated (Section 3.2) and then evaluated against available satellite and in situ data (Sections 3.3–3.5). Analysis of the results show how each emission scheme affects the IFS-AER performance (Section 4.1). We also discuss the use of satellite-based data and parameterization of whitecap fraction (Section 4.2) and the effects of the scaling and shape factors on the IFS-AER products (Section 4.3). We conclude with the main points of this work and a look forward (Section 5).

## 2. Models and Data

### 2.1. Main Characteristics of IFS-AER

The IFS-AER uses mass mixing ratio as the prognostic variable of the aerosol tracers, an approach derived from the general circulation model LOA/LMDZ of the Laboratoire d'Optique Atmosphérique (LOA) and the Laboratoire de Météorologie Dynamique (LMD) [12,13]. Starting with the implementation of operational cycle 46R1 in July 2019, the prognostic species are sea salt, desert dust, organic matter, black carbon, sulfate, nitrate, and ammonium. By default, the IFS-AER runs are coupled with an operational global chemistry scheme, which is an extended version of the so-called Carbon Bond Mechanism 5 (CB05) [14] that has been integrated into the IFS [9,15] and is denoted as "IFS-CB05". However, the IFS-AER can also run in stand-alone mode, i.e., without any interaction with the chemistry scheme. In this case, the nitrate and ammonium species are not included and a specific tracer representing sulfur dioxide is added as described in [5]. Since October 2021, the IFS-AER operational version is cycle 47R3. More details about the operational IFS-AER versions can be found at https://confluence.ecmwf.int/display/COPSRV/CAMS+Global (Last accessed on 17 September 2021). The operational cycle 47R1 IFS is described in [16].

Sea salt aerosols are represented with three bins, fine (bin 1), coarse (bin 2), and super coarse (bin 3), with radius bin limits at 0.03, 0.5, 5, and 20 μm. Sea salt particle radii and emissions are expressed at RH of 80% [13]. This is different from all the other aerosol species in the IFS-AER, which are expressed as dry mixing ratios (RH = 0%). Therefore,

the sea salt aerosol mass mixing ratio needs to be divided by a factor of 4.3 to convert to dry mass mixing ratio in order to account for the hygroscopic growth and change in density. The 4.3 factor corresponds to the ratio of the sea salt assumed volume and density at RH of 80 and 0%. Desert dust is also represented with three bins (radius bin limits are 0.03, 0.55, 0.9, and 20 µm). For both dust and sea salt, there is no mass transfer between bins. For organic matter and black carbon, two components are considered: hydrophilic and hydrophobic fractions, with the aging processes transferring mass from the hydrophobic to hydrophilic organic matter and black carbon. Sulfate aerosols—as well as their precursor gas sulfur dioxide, used when the IFS-AER is run not fully coupled with IFS-CB05—are represented by one prognostic variable each. When running the fully coupled scheme IFS-CB05, sulfur dioxide is presented in the IFS-CB05 module and is thus absent in the IFS-AER. Nitrate and ammonium are two extra species, first included in the operational IFS-AER products in cycle 46R1. For the nitrate species, two prognostic variables represent fine nitrate produced by gas–particle partitioning and coarse nitrate produced by heterogeneous reactions on or within dust and sea salt particles.

In all, the IFS-AER is composed of 12 prognostic variables when running stand-alone and 14 when fully coupled with IFS-CB05 (including nitrates and ammonium). These operational configurations allow for a relatively limited consumption of computing resources.

### 2.2. Sea Salt Aerosol Emission Schemes in the IFS-AER

Most sea salt emission schemes evaluated for use in the IFS-AER employ SSSFs based on the whitecap method (Section 1). Therefore, a parameterization of the whitecap fraction is usually required to estimate sea salt emissions. Traditionally, whitecap fraction is parameterized as a function of wind speed at 10 m reference height $W(U_{10})$. However, in situ field measurements of sea spray production do not constrain well sea salt emissions predicted using the relationship $W(U_{10})$ alone [11,17]. Other variables, besides the wind, affect the formation of whitecaps and the sea spray production [18]. Thus, it is necessary to use a $W$ parameterization that better accounts for additional meteorological and oceanographic (MetOc) effects. The sea salt emission schemes used in the IFS-AER fulfill this need with different approaches.

The size distributions in the shape factor of the SSSF have been presented using various particles sizes, from radius of formation $r_0$, to radius at 80% RH $r_{80}$, to dry particle radius $r_d$. Conversions between different particle sizes require respective conversions of the SSSFs [19]. For consistency in notations, we use the relationship $r_{80} \cong 2r_d \cong D_p$ (where $D_p$ is the dry particle diameter) to present $dF/dr_{80} \cong dF/dD_p$.

#### 2.2.1. First Sea Salt Emission Scheme: Monahan et al. (1986)

The SSSF formulated by Monahan et al. ([20], hereafter M86) is the first one implemented in the IFS-AER. It is formulated as follows:

$$\frac{dF}{dD_p} = W(U_{10}) \times 3.5755 \times 10^5 D_p^{-3}(1 + 0.057D_p^{1.05}) \times 10^{1.19\exp(-B^2)} \tag{2}$$

where

$$B = \frac{0.38 - \log(D_p)}{0.65} \tag{3}$$

The shape function given with (2) and (3) is applicable for $D_p$ = 0.8–8 µm. Therefore, M86 parameterization scheme is outside of its range of applicability when used to simulate fine (bin 1) and super coarse (bin 3) sea salt particles (Section 2.1).

The scaling factor in M86 is the whitecap fraction parameterized as a function of wind speed only using the power law formulation proposed by [21]:

$$W(U_{10}) = 3.84 \times 10^{-6} U_{10}^{3.41} \tag{4}$$

Sea salt emissions obtained with Equations (2)–(4) have been used in many models (see Table 1 in [11]). Hereafter, we refer to parameterization scheme (2)–(4) as M86.

2.2.2. Recent Sea Salt Emission Scheme: Grythe et al. (2014)

The more recent parameterization of Grythe et al. ([19], hereafter G14) has been implemented in operational cycle 45R1 of the IFS-AER. It combines emissions at three size modes centered at $D_p$ of 0.1, 3, and 30 µm. Whitecap fraction is not used in G14 SSSF formulation, and thus the wind speed dependence is introduced in a different manner—via the shape function, which has different power laws, developed by statistical fits to observations, for each size mode:

$$\frac{dF}{dD_p} = T_W(T) \left( 235\, U_{10}^{3.5} \exp\left( -0.55(\ln \frac{D_p}{0.1})\right)^2 \right)$$

$$+ T_W(T) \left( 0.2\, U_{10}^{3.5} \exp\left( -1.5(\ln \frac{D_p}{3})\right)^2 \right)$$

$$+ T_W(T) \left( 6.8\, U_{10}^{3} \exp\left( -(\ln \frac{D_p}{30})\right)^2 \right) \tag{5}$$

Considering the size mode centers in Equation (5) and the IFS-AER bins (Section 2.1), G14 is used within its range of applicability.

The scaling factor in G14 introduces dependence on sea surface temperature (SST) $T$ with function $T_W(T)$. Its form is as developed by [17]:

$$T_W(T) = 0.3 + 0.1\, T - 0.0076\, T^2 + 0.00021\, T^3 \tag{6}$$

Factor $T_W(T)$ yields an increase in sea salt emissions over the tropics and regions with warmer waters. Such an SST effect is consistent with the conclusions that modeled marine aerosol optical depths (AOD) are generally too low in the tropics [17,22]. Because of the scarcity and heterogeneity of the observational data, there are large uncertainties in the temperature dependence of sea salt aerosol production [19].

Hereafter, we refer to parameterization scheme (5) and (6) as G14.

2.2.3. New Sea Salt Emission Schemes: Albert et al. (2016)

The new sea salt emission scheme implemented in the operational cycle 47R1 of the IFS-AER is based on the whitecap method (Section 1) and thus requires to first estimate the whitecap fraction. For this, we use the parameterization of Albert et al. ([23], hereafter A16), which quantifies the whitecap fraction as a function of both wind speed and SST, $W(U_{10}, T)$. In contrast to $W(U_{10})$ in M86, developed by [21] on the basis of in situ $W$ observations, Ref. [23] used $W$ values from passive remote sensing observations of the ocean surface by the WindSat microwave radiometer (more in Section 2.3.1). These satellite-based $W$ values are available at two frequencies, 10 and 37 GHz, which give some sense for the foam thickness, thus the stage of the whitecap lifetime—active or decaying [23]. Active (young) whitecaps comprise thick foam layers sensed well by 10 GHz, while 37 GHz detects the full distribution of foam thicknesses, from thick to thin, and thus accounting for both active and decaying (mature) whitecaps.

To develop the $W(U_{10}, T)$ relationship, Ref. [23] used a database of whitecap fraction and time-space matched MetOc variables for 2006. Wind speed $U_{10}$ was from the SeaWinds scatterometer on the QuikSCAT platform and SST $T$ was from the Global Data Assimilation System (GDAS) at the U.S. National Center for Environmental Prediction (NCEP). See [23] for extensive details on the data used and the parameterization methodology. Because parameterization $W(U_{10}, T)$ was developed from data spanning the globe, as opposed to the location-specific in situ data used by [21] for $W(U_{10})$ in (4), it is conceivable that the new

relationship better represents the whitecap fraction on a global scale [24]. The functional form of the new parameterization $W(U_{10}, T)$ is:

$$W = a(T) [U_{10} + b(T)]^2 \tag{7}$$

where

$$\begin{aligned} a(T) &= a_0 + a_1 T + a_2 T^2 \\ b(T) &= b_0 + b_1 T \end{aligned} \tag{8}$$

Table 5 in [23] lists the fitting coefficients $a_{0,1,2}$ and $b_{0,1}$ for each frequency, 10 and 37 GHz. In the IFS-AER implementation of this scheme, using the fit for 37 GHz gave better results, thus coefficients $a_{0,1,2}$ and $b_{0,1}$ for this frequency were chosen for the implementation of the new scheme.

Figure 1 compares the simulated whitecap fraction using parameterizations (4) from [21] (M80 in the figure) and (7) and (8) from [23] (A16 in the figure). The SST dependence of parameterization (7) and (8) contrasts the lack of it in (4), while the higher power law used in (4) gives a sharper gradient as a function of 10 m wind speed.

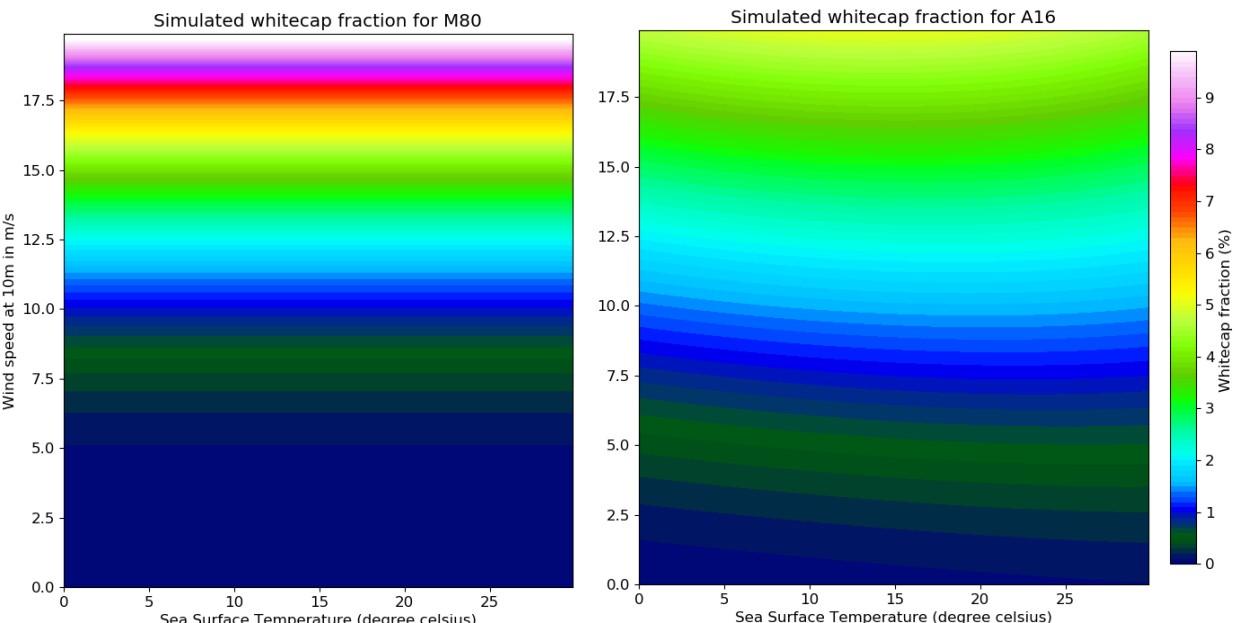

**Figure 1.** Comparison of simulated whitecap fraction as a function of 10 m wind speed and SST between the M80 (**left**) and A16 (**right**) parameterizations.

Sea salt emission is obtained by combining parameterization A16 for the whitecap fraction with the sea spray size distribution (shape factor) represented either as (2) and (3) or its modification by [25]. The modified form is:

$$\frac{dF}{dD_p} = W \times 3.5755 \times 10^5 D_p^{-A} (1 + 0.057 D_p^{3.45}) \times 10^{1.607 \exp(-B^2)} \tag{9}$$

where

$$\begin{aligned} A &= 4.7(1 + \theta D_p)^{-0.017 D_p^{-1.44}} \\ B &= \frac{0.433 - \log(D_p)}{0.433} \end{aligned} \tag{10}$$

Here, $\theta$ is an adjustable parameter, which [25] introduced to fit the SSSF values to field observations of [26]. It provides additional control of the shape of the sub-micron size

distribution of the generated sea spray. Following the original parameterization of [25], we set $\theta = 30$. The modification aimed to extend the range of size distribution (2) and (3). The size distribution given with (9) and (10) is applicable for $D_p = 0.07$–20 µm, almost fully covered by bins 1–3 (Section 2.1).

In this work, new sea salt emissions are obtained using (7) and (8) with $U_{10}$ and $T$ from ECMWF in combination with the size distributions (2) and (3) (parameterization scheme A16) and (9) and (10) (parameterization scheme A16E). As of October 2020, emission scheme A16 is included in operational cycle 47R1 of the IFS-AER [16].

### 2.3. Data for Evaluating Sea Salt Emission Schemes

The IFS-AER computes various characteristics of the sea salt aerosol using the sea salt emissions obtained with parameterization schemes M86, G14, A16, and A16E. These characteristics are burden, surface concentration, lifetime, and AOD. In this study, we evaluate the skill of each parameterization scheme by comparing the simulated sea salt aerosol characteristics to satellite or in situ observations. Brief account of the data used for comparison and evaluation follows.

#### 2.3.1. Whitecap Fraction

We use satellite-based observations of whitecap fraction instead of in situ $W$ data for three reasons. First, the available in situ $W$ data are insufficient for evaluating the IFS-AER simulations. About 1500 data points of whitecap fraction—extracted from sea surface photographs taken on towers and ships—have been collected since the 1960s (see Table 2 in [18] for data sets up to 2004 and Table 1 in [27] for data sets after 2004). Next, the $W$ values have been obtained with image processing algorithms, which, while based on the same intensity threshold method, have been implemented differently for each individual data set. This inconsistency contributes to the in situ $W$ data spread, which is already large due to both measuring error and geophysical variability. Finally, the $W$ values are reported with respective $U_{10}$ values, but no auxiliary information, such as time and location (latitude, longitude), is available to match them in time and space to the IFS-AER results. Of course, the satellite-based $W$ values also contain measuring error and geophysical variability. However, the retrieval method is consistent for all data points and a large amount of data are available with information for time-space matching to the IFS-AER results.

We use satellite $W$ retrievals for 2014 obtained from WindSat observations of brightness temperature $T_B$ using the latest version of the whitecap retrieval algorithm $W(T_B)$ [27]. WindSat retrievals of wind vector (speed and direction), SST, water vapors, and cloud liquid water are used as inputs to the models comprising the $W(T_B)$ algorithm, namely: atmospheric model for atmospheric correction, dielectric constant model for the specular (flat surface) emissivity, 2-scale and wave spectrum models for the emissivity of rough surface, and foam emissivity model. Whitecap fraction retrievals are available at three microwave frequencies of 10, 18, and 37 GHz, horizontal (H) and vertical (V) polarizations. All WindSat data are at high resolution (footprint 25 by 35 km). The daily $W$ retrievals are subsequently gridded at spacial resolution of $1/4° \times 1/4°$ latitude–longitude grid cell.

The $W$ data for 2006, used by [23] to derive the A16 parameterization (7) and (8), are from an earlier version of the whitecap algorithm. The major differences between the algorithms producing $W$ data for 2006 and 2014 include atmospheric correction modeling—simplified in 2006, full model in 2014; source of forcing variables—external data for 2006, WindSat geophysical retrievals for 2014; raw (swath) data resolution—low (large satellite footprint) in 2006, high in 2014; and gridding resolution—0.5° in 2006, 0.25° in 2014. Please refer to [27] for extensive details.

#### 2.3.2. Surface Concentration of Sea Salt Aerosol

Surface concentration of sea salt aerosols (denoted SS) is obtained by combining measured surface concentrations of sodium and chlorine ions, $Na^+$ and $Cl^-$ (see [17] and

the references therein, as well as [28]). Typically, aerosols are collected by high-volume filter samplers with PM10 inlets (i.e., particles less than 10 μm in diameter) and analyzed for the major aerosol species. Assuming that all measured $Na^+$ and $Cl^-$ concentrations are derived from seawater, sea salt concentrations are calculated as:

$$[\text{SS}] = 1.47[\text{Na}^+] + [\text{Cl}^-] \tag{11}$$

We use two sources for measurements of $Na^+$ and $Cl^-$ concentrations. First, we use historical observations (as monthly climatologies) from the AEROCE (Atmosphere/Ocean Chemistry Experiment) and SEAREX (Sea/Air Exchange) programs, carried out by the University of Miami in the 1980s and 1990s. Figure 2 shows the stations used in this study (20 out of 35 in total) for the SS climatological evaluation. Because these observations were conducted in ambient conditions with PM10 inlets [17], they provide approximately (with possible slight overestimation) the sum of SS in bin 1 and bin 2 (i.e., sea salt aerosols with $r_{80} \leq 5$ μm).

The second source for surface concentration of $Na^+$ and $Cl^-$ is the Clean Air Status and Trends Network (CASTNET), comprising more than 90 stations over the continental United States (www.epa.gov/castnet, last accessed on 8 October 2021). The CASTNET observations are made with a filter sampler without any size cutoff (i.e., measured is the the total amount of particles suspended in the atmosphere). This means that some super large sea salt aerosol particles (with $r_0 > 20$ μm) could be observed but are not simulated with the bins used in the IFS-AER (Section 2.1). While this likely affects data from stations in coastal cities, it is not likely to impact results significantly because of the very rapid deposition of these particles via gravitational settling. We use weekly observations for 2014 from all CASTNET sites.

Because the CASTNET sites are continental, the simulated SS depends at least as much on sinks (sedimentation, wet and dry deposition) as on emission sources. While this source–sink balance may reduce the significance of the CASTNET observations for evaluating the sea salt aerosol schemes, it is notable that sea salt particles could be transported up to hundreds of kilometers inland [28]. Furthermore, the observed surface concentration of $NA^+$ at CASTNET sites also includes contributions from other sources, such as mineral dust and anthropogenic. However, the contribution of anthropogenic sources to $Na^+$ is generally low [29], and the effect of mineral dust sources is localized over the U.S. Considering all these, it was decided to include this evaluation because (i) the emissions of sea salt aerosol affect the continental sites via the transport and sink of sea salt and (ii) the simulation of speciated aerosol surface concentration is one of the key products of CAMS.

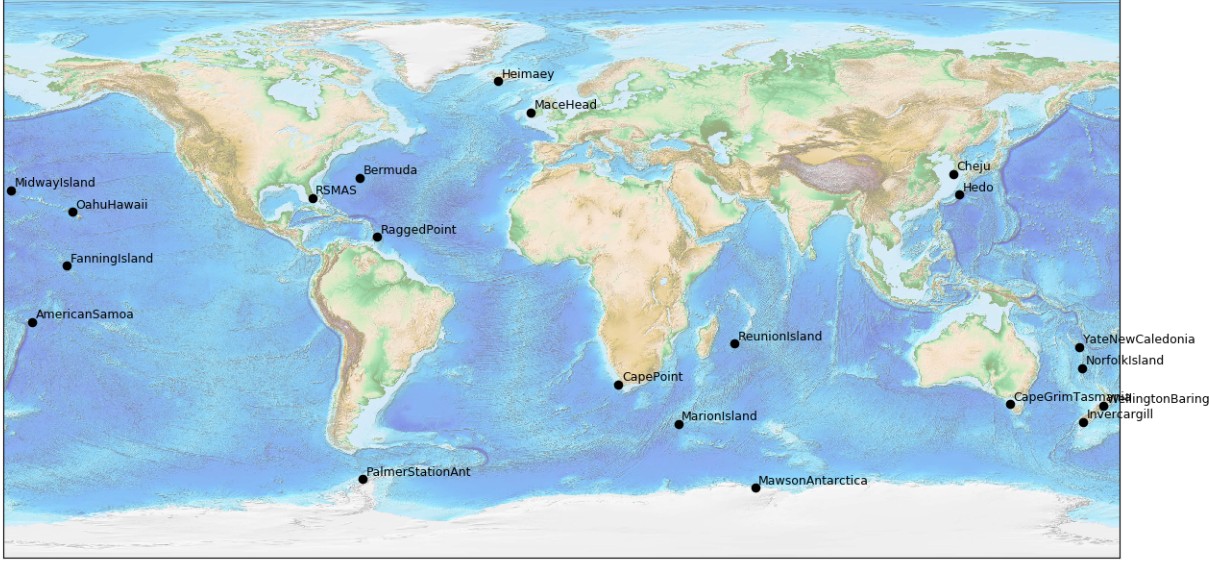

**Figure 2.** Stations from the AEROCE/SEAREX networks used in this study.

### 2.3.3. Aerosol Optical Depth

At each model level, extinction caused by sea salt aerosol is computed by multiplying the prognostic sea salt aerosol mass mixing ratio with the mass extinction for each of the three bins. The mass extinction from sea salt aerosol has been estimated for each of the three bins using the Mie theory, as described in [30], assuming spherical shapes for aerosol types. We obtain sea salt AOD by integrating over the vertical extinction profile. In this study, we use sea salt AOD data at 550 nm summed over the three bins. This process is also performed for the other aerosol species of the IFS-AER, and the total AOD is computed as the sum of the AOD of all species, using an external mixing assumption.

We evaluate the simulated AOD with both satellite and in situ data for AOD. The Finnish Meteorological Institute (FMI) provides a merged AOD product at 550 nm as gridded ($1° × 1°$) monthly AOD data over long period (from 1995 to 2017). This product has been constructed by merging retrievals of AOD from various satellite sensors; the merging follows different approaches. The choice of the merged product is based on comparisons to in situ observations of AOD from the Aerosol Robotic Network (AERONET, Holben et al., 1998 [31]). Extensive details of the FMI merged AOD are given in [32]. Over most of oceans, the AOD from FMI is representative of the sea salt AOD because other aerosol species have only a local/regional impact on average, such as dust over the Atlantic off the west coast of Africa and anthropogenic aerosols over the Western Pacific.

We also compare our simulated AOD directly to AERONET observations of AOD at 500 nm at 14 selected sites that are representative of sea salt aerosol in open ocean or coastal zone (Holben et al., 1998 [31]). The AERONET stations used are: Ragged Point, Reunion St Denis, Noumea, Midway Island, Key Biscayne, Key Biscayne2, Cape San Juan, Edinburgh, Cabo da Roca, ARM Graciosa, American Samoa, Amsterdam Island, Andenes, and Birkenes. AERONET observations give the total AOD, i.e., the AOD of all aerosol species, and is compared to the simulated AOD at 500 nm, which is interpolated to the AERONET sites. We then apply a time averaging to obtain weekly skill scores.

We use two metrics to assess the simulated AOD against observations, namely modified normalized mean bias (MNMB) and fractional gross error (FGE). The MNMB varies between $-2$ and 2. For a population of $N$ forecasts $f_i$ and observations $o_i$, MNMB is defined as:

$$\text{MNMB} = \frac{2}{N} \sum_i \frac{f_i - o_i}{f_i + o_i} \tag{12}$$

The FGE varies between 0 (best) and 2 (worst), and is defined for a population of N forecasts $f_i$ and observations $o_i$ as:

$$\text{FGE} = \frac{2}{N} \sum_i \left| \frac{f_i - o_i}{f_i + o_i} \right| \tag{13}$$

The use of MNMB and FGE allows the investigator to focus more on the skill of the model in simulating relatively low AOD values. Sea salt AOD is typically lower than the AOD of other aerosol types because of the lower mass extinction of sea salt aerosol. This is so for two reasons: (i) sea salt particles are generally larger in size than other species, and thus they are less light scattering for the same mass of finer aerosols; and (ii) due to their hygroscopicity, sea salt particles have higher water content than most other aerosol species.

## 3. Results: Simulations and Evaluation

### 3.1. Experiments

Four experiments have been carried out to test and evaluate the two existing sea salt emission schemes as well as the two newly developed schemes. The experiments are named to the respective parameterization schemes they use (Sections 2.2.1–2.2.3). Table 1

lists these experiments (along with their IFS-AER operational cycles) and summarizes the main elements comprising each of them.

**Table 1.** Main elements of IFS-AER simulation experiments.

| Experiment | Operational Cycle | Scaling Factor | Shape Function | Size Range ($D_p$ µm) |
|---|---|---|---|---|
| M86 | 43R3 | Equation (4) | Equations (2) and (3) | 0.8–8 |
| G14 | 45R1, 46R1 | Equation (6) | Equation (5) | 0.1–30 |
| A16 | 47R1 | Equations (7) and (8) | Equations (2) and (3) | 0.8–8 |
| A16E | NA | Equations (7) and (8) | Equations (9) and (10) | 0.07–20 |

IFS-AER cycle 47R1 has been run in forecast only mode (i.e., without any data assimilation) to simulate the year 2014, with fifteen days of spinup time. The resolution used is about 80 km grid size, with 137 levels over the vertical. Apart from the horizontal resolution, the configuration of the simulations is similar to those described in [16], using the more recent dry and wet deposition parameterizations (derived from [33] for wet deposition, following [34] for dry deposition). For experiments M86, A16, and A16E, the simulated whitecap fraction has been archived so as to be compared against retrievals. No whitecap fraction is explicitly simulated for G14 (Section 2.2.2).

*3.2. Budgets*

Table 2 shows the simulated emissions, burden, and lifetime for each experiment in the three sea salt bins (Section 2.1). Experiment G14 simulates the highest sea salt emissions (by size and total), mostly via the super coarse sea salt aerosol (bin 3). Experiment A16 yields the lowest particle lifetime of the coarse (bin 2) and super coarse (bin 3) sizes.

With a fixed shape function (2) and (3), the sea salt emissions increase 2–3 fold for all size bins when using scaling factor (7) and (8) in experiment A16 instead of scaling factor (4) in experiment M86. However, the increase brought about by scaling factor (7) and (8) in experiments A16 and A16E is constrained by shape function (9) and (10) used in A16E, especially for the coarse and super coarse sea salt aerosol. The reason is that the differences in both shape and range of the size distributions of experiments A16 and A16E affect the balance between emission and deposition of sea salt particles.

**Table 2.** Dry sea salt aerosol emissions, burden, and lifetime simulated by IFS-AER with four emission schemes (Table 1). The emissions are in Tg·yr$^{-1}$, the burdens are in Tg, and the lifetimes are in days.

| Process | Bin 1 | Bin 2 | Bin 3 | Total |
|---|---|---|---|---|
| Emissions (M86) | 32.2 | 2767.2 | 3363.8 | 6163.2 |
| Burden (M86) | 0.09 | 3.53 | 1.43 | 5.05 |
| Lifetime (M86) | 1.0 | 0.46 | 0.16 | 0.29 |
| Emissions (G14) | 41.6 | 1799.5 | 45,531.6 | 47,372.7 |
| Burden (G14) | 0.14 | 2.86 | 22.5 | 22.5 |
| Lifetime (G14) | 1.3 | 0.58 | 0.18 | 0.2 |
| Emissions (A16) | 110.3 | 6595.5 | 13,657.8 | 20,363.6 |
| Burden (A16) | 0.39 | 4.46 | 1.41 | 6.2 |
| Lifetime (A16) | 1.3 | 0.25 | 0.04 | 0.11 |
| Emissions (A16E) | 197.8 | 2444.6 | 9190.3 | 11,832.7 |
| Burden (A16E) | 0.7 | 3.45 | 1.87 | 6.02 |
| Lifetime (A16E) | 1.29 | 0.52 | 0.07 | 0.19 |

*3.3. Evaluation of the Simulated Whitecap Fraction*

In this section, the annual whitecap fraction values simulated by experiments M86 and A16 are compared against retrievals of whitecap fraction from WindSat (Section 2.3.1). Figure 3 shows the retrieved and simulated whitecap fraction values, formed by collocating

daily *W* data and subsequently averaging for 2014. The retrieved whitecap fraction varies between 1–2.5%, on average, in the high latitudes, and 0.3 to 1% in the tropics and around the equator. Values from M86 are much lower than the retrievals between 40° S and N. The average values from A16 are closer to the retrievals, with higher values in the Southern Ocean.

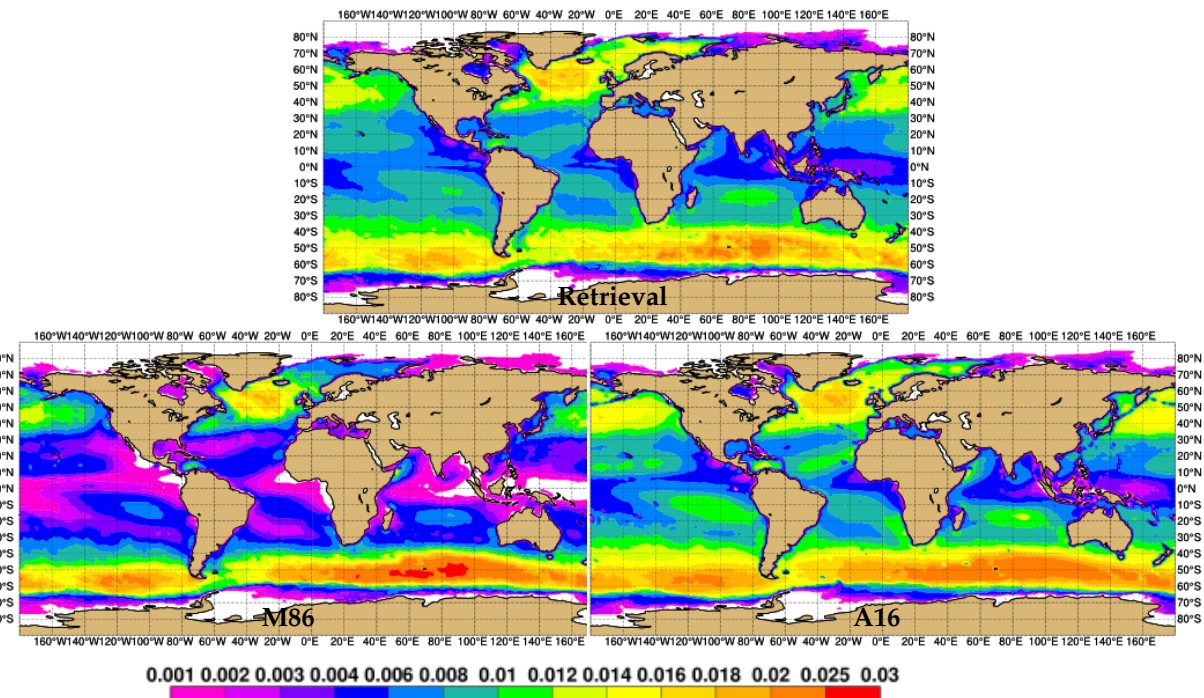

**Figure 3.** (**Top**): 2014 average of retrieved whitecap fraction. (**Bottom**): collocated 2014 average of whitecap fraction simulated by M86 (**left**) and A16 (**right**).

Figure 4 shows the mean bias and the root-mean-square error (RMSE) of simulated whitecap fraction as compared to retrievals. These statistical metrics allow refined comparison of observed and simulated *W* values. Relationship (4), used in experiment M86, is biased low (by up to −0.6%) from the retrievals almost everywhere and is higher (by up to +0.6%) in regions with high wind speed (top left in Figure 4). This is due to the higher wind speed exponent—(3.41) in (4)—as compared to the quadratic expression (7) used in A16. Experiment A16 shows a small positive bias (up to +0.2%) almost everywhere, except at the equator where local negative bias of −0.2% occurs (bottom left in Figure 4). Note that this bias is most probably due to the use of different algorithm versions for the *W* estimates for 2006, on which (7) was derived, and the retrievals for 2014 (Section 2.3.1). The RMSE shows a similar pattern (right column in Figure 4): the deviations from the retrievals are significantly higher for (4) in M86 (between 0.2 and 1%) as compared to A16. With A16, the RMSE is 0.1 to 0.5%.

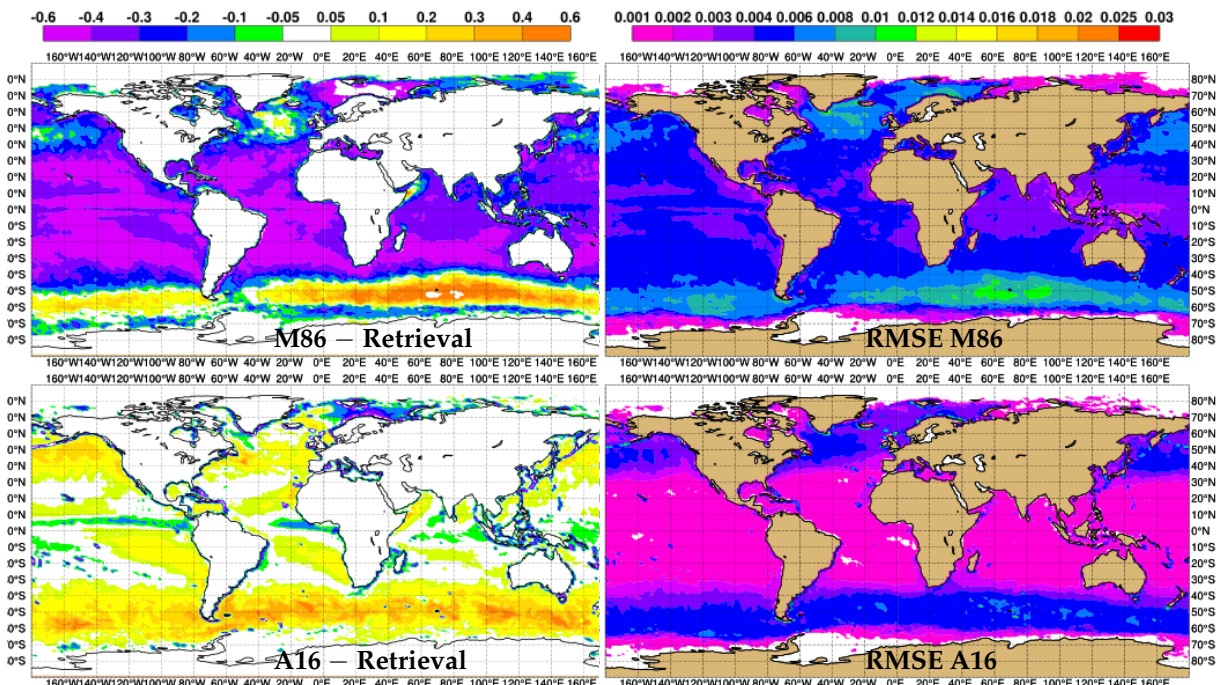

**Figure 4.** 2014 average bias (**left**) and RMSE (**right**) of simulated daily whitecap fraction as compared to collocated daily retrievals for experiments M86 (**top**) and A16 (**bottom**).

### 3.4. Evaluation of the Simulated Sea Salt Aerosol Surface Concentration

The results of SS evaluation with AEROCE/SEAREX observations are shown in Figure 5; the sea salt emissions of fine and coarse particles (bins 1 and 2) are used for the comparison (recall Section 2.3.2). The figure shows density scatter plots of observation (obs) values versus modeled (mod) values for each experiment (Table 1), together with distributions for each set of data (obs and mod) and statistics (bias, RMSE, and correlation coefficient *r*). The dotted red lines indicate the area where the difference between simulated and retrieved values is less then 2 $\mu g/m^3$. A *t*-test has been carried out for each experiment; the results showed that the differences between observed and simulated values are statistically significant for all experiments.

Figure 5 shows that all experiments underestimate the monthly SS as compared to AEROCE/SEAREX climatological observations. This underestimation is the worst for G14 (bias of $-9.4$ $\mu g/m^3$), because of relatively lower emissions of sea salt aerosol in bins 1 and 2. With the G14 scheme, the emissions of super coarse sea salt aerosol (bin 3) are much larger than with the other emission schemes (Table 2), but this has no effect here, as the sum of sea salt aerosol mass mixing ratio of bins 1 and 2 are used in this evaluation. The metrics for experiment M86 are close to those of G14. The best performance is that of experiment A16 with the lowest bias ($-3.4$ $\mu g/m^3$) and RMSE. The evaluation metrics deteriorate for experiment A16E compared to A16, but they still improve upon experiments M86 and G14.

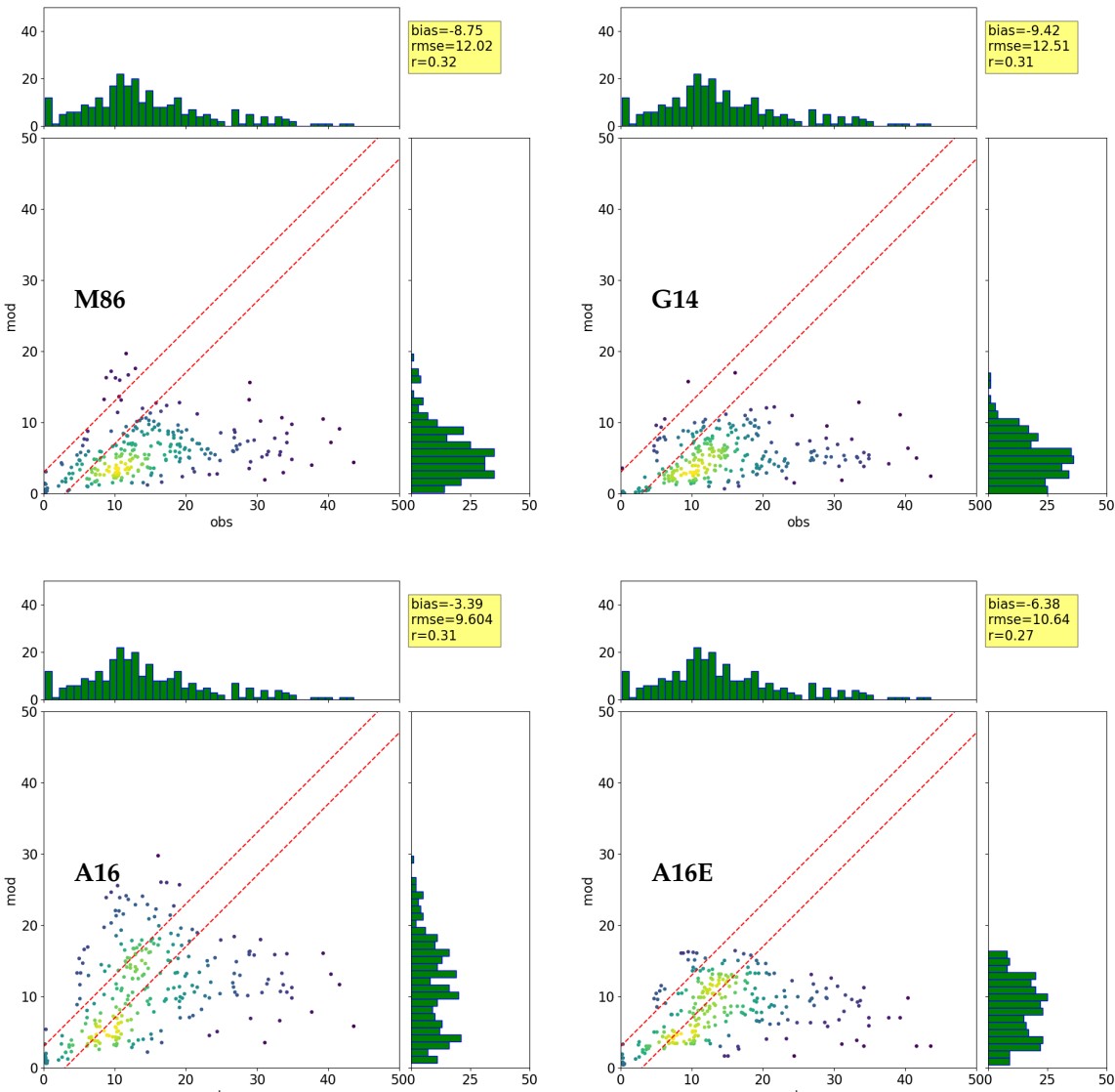

**Figure 5.** Density scatter plots of simulated monthly sea salt aerosol surface concentration (in μg/m$^3$, PM10 size range) against climatological values from the AEROCE/SEAREX observations

Figure 6 shows the results of SS evaluation over all CASTNET sites in the same format as in Figure 5; the sea salt emissions in all bins (from fine to super coarse) are used in this case (Section 2.3.2). The simulated SS for M86 mostly underestimates the observations, while G14 mostly overestimates them. Both A16 and A16E cluster the obs–mod pairs closer to the 1:1 line, slightly better for A16E than for A16. The existing experiments (M86 and G14) show lower correlation coefficients compared to the new A16 and A16E experiments.

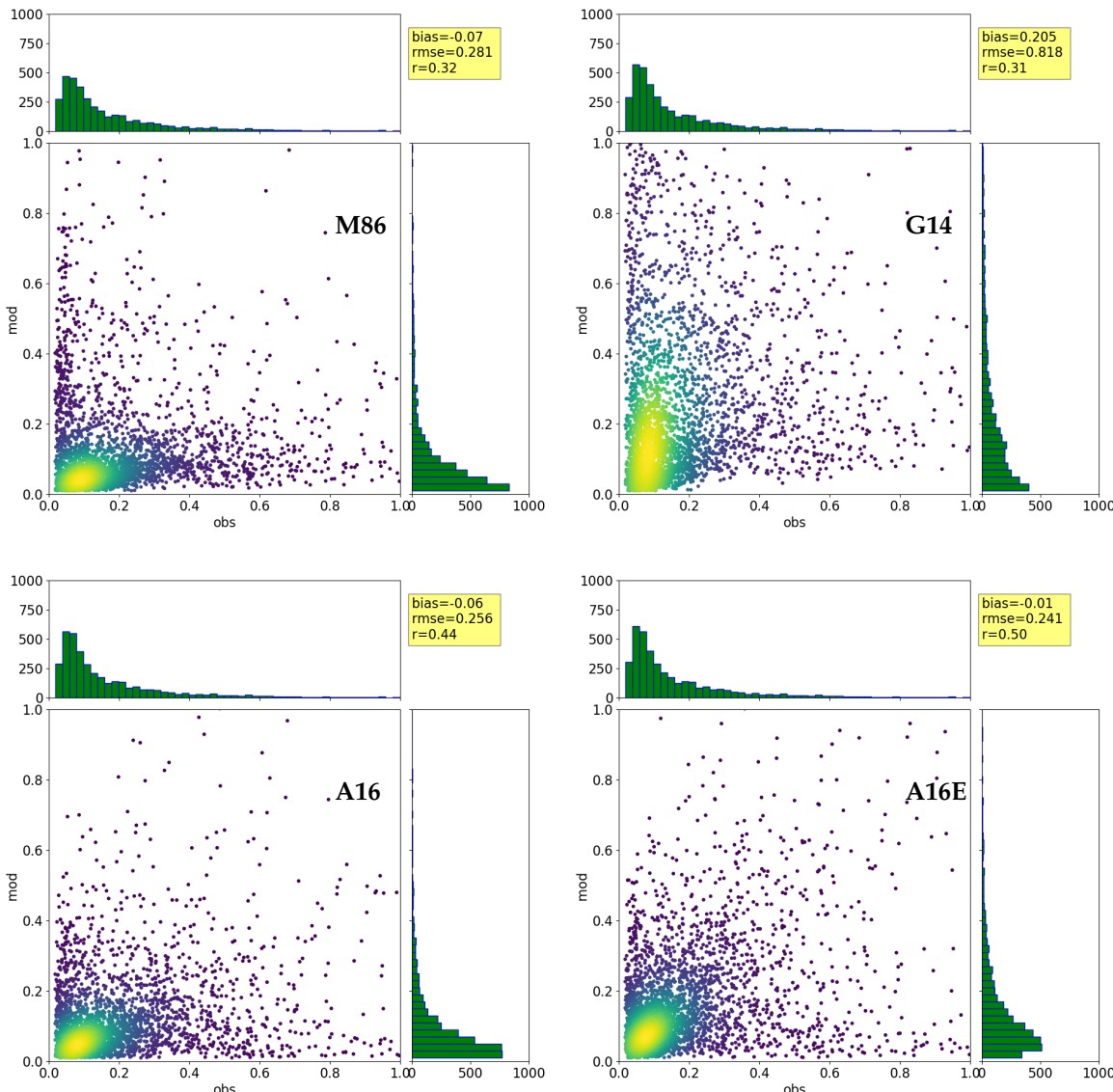

**Figure 6.** Density scatter plots of simulated weekly sea salt aerosol surface concentration (in µg/m$^3$) against values from the CASTNET network.

*3.5. Evaluation of the Simulated Aerosol Optical Depth*

It has been shown previously that the simulated sea salt AOD at 550 nm is much higher with G14 as compared to M86 [5]. Figure 7 shows that the sea salt AOD simulated by A16 increases the AOD above that of the M86 experiment, but is much lower than the AOD from the G14 experiment. Experiment A16E further increases the AOD, but it is still lower than the G14 simulation. Both A16 and A16E experiments simulate less marked low and high values. This reflects the fact that whitecap fraction (and the resulting sea salt AOD) are estimated with quadratic wind speed power law in A16 and power 3 to 3.5 with G14.

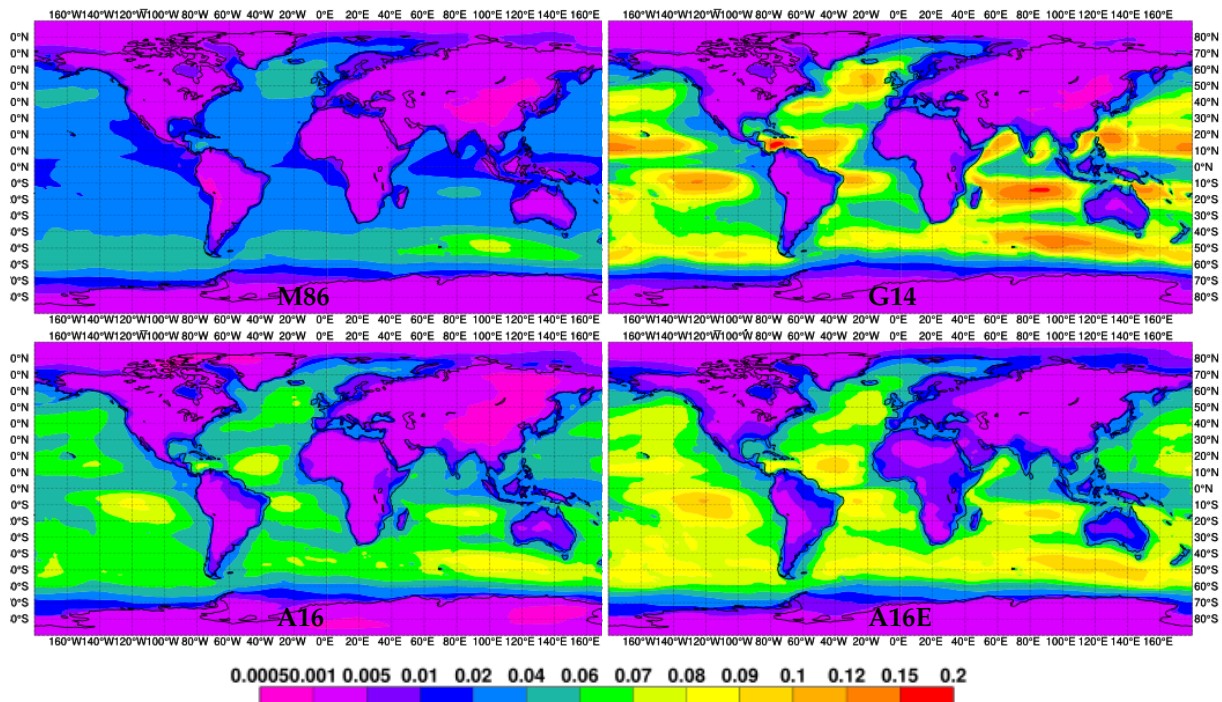

**Figure 7.** Sea salt AOD at 550 nm, averaged for 2014, as simulated by M86 (**top left**), G14 (**top right**), A16 (**bottom left**), and A16E (**bottom right**).

Figure 8 shows a comparison of MNMB (left panels) and FGE (right panels) of the simulated total AOD to the FMI merged product (Section 2.3.3). The changes in the MNMB of AOD indicate improved total AOD over most of the oceans (where sea salt aerosol generally dominates over other species) by G14, A16, and A16E experiments (more blue colors) as compared to M86, which is characterized by a significant low bias (more red colors). The FGE changes towards lower values (i.e., improving total AOD values) with G14 as compared to M86: FGE is between 0.1 and 0.2 over most of the oceans for G14, against values of 0.3–0.7 for M86. The FGE is also improved by A16 (values from 0.1–0.2 over most of the oceans), and further improved by A16E (with values between 0.05 and 0.15 ) over nearly all oceans. Table 3 summarizes the mean global value of MNMB and FGE for all experiments and illustrates the significant improvement in the skill of simulated AOD at 550 nm brought by the two A16 and A16E experiments.

Figure 9 compares the simulated total AOD from all experiments with the AERONET data (top panel). The bottom panels show the MNMB (left) and the FGE (right) values. We see that G14, A16, and A16E significantly reduce the low bias of M86 as compared to AERONET. Experiment A16 improves on the FGE as compared to G14 and more so against M86. Similar changes occur for experiment A16E, which shows the highest skill.

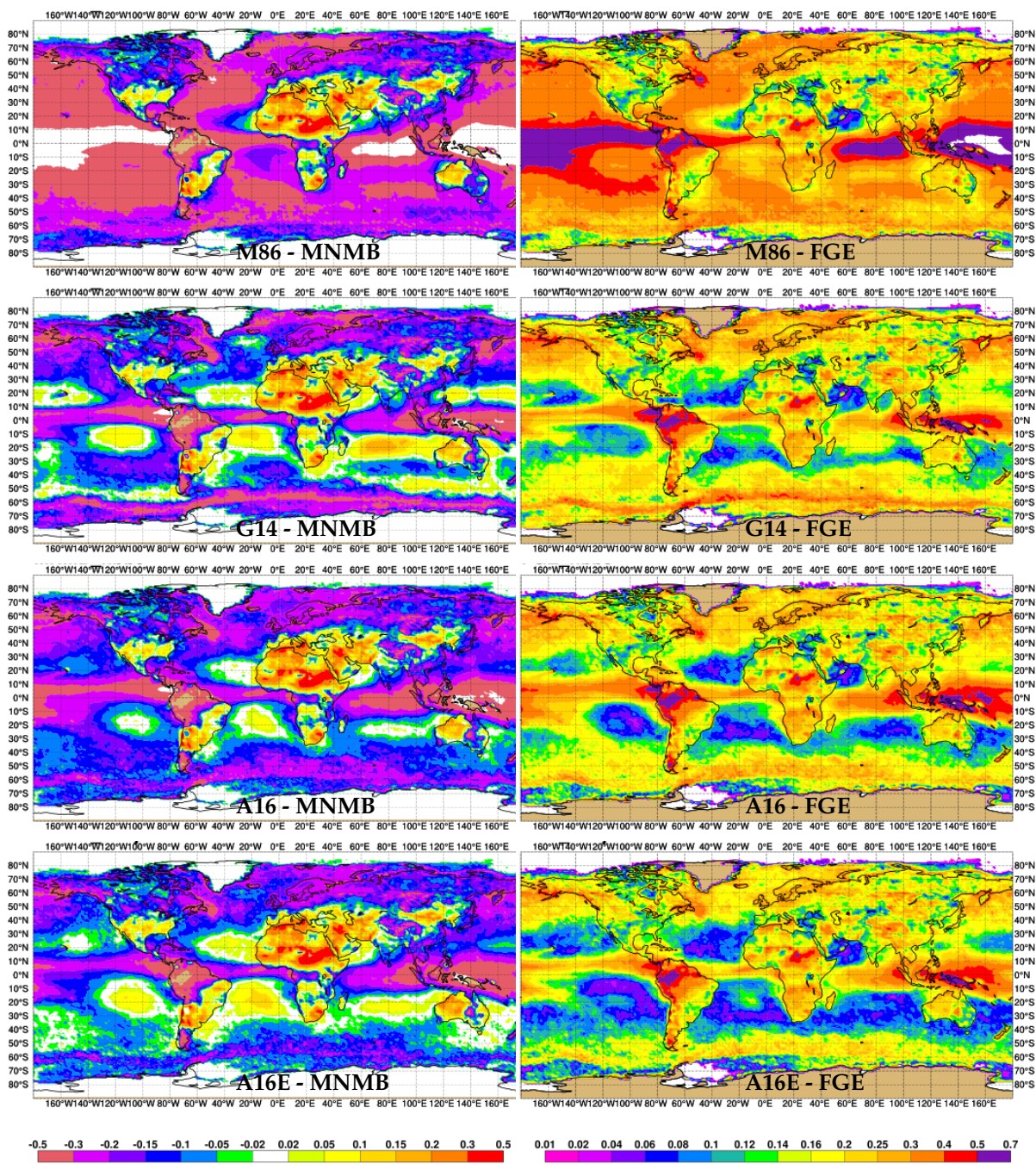

**Figure 8.** Simulated total AOD at 550 nm against the merged AOD product from FMI as evaluated with (**left**) MNMB and (**right**) FGE of M86 (**first row**), G14 (**second row**), A16 (**third row**), and A16E (**bottom**).

**Table 3.** Mean global MNMB and FGE of total AOD at 550 nm simulated by the four IFS-AER experiments against the merged AOD product from [32].

| Experiment | MNMB | FGE |
|---|---|---|
| M86 | −0.207 | 0.241 |
| G14 | −0.099 | 0.168 |
| A16 | −0.119 | 0.164 |
| A16E | −0.084 | 0.142 |

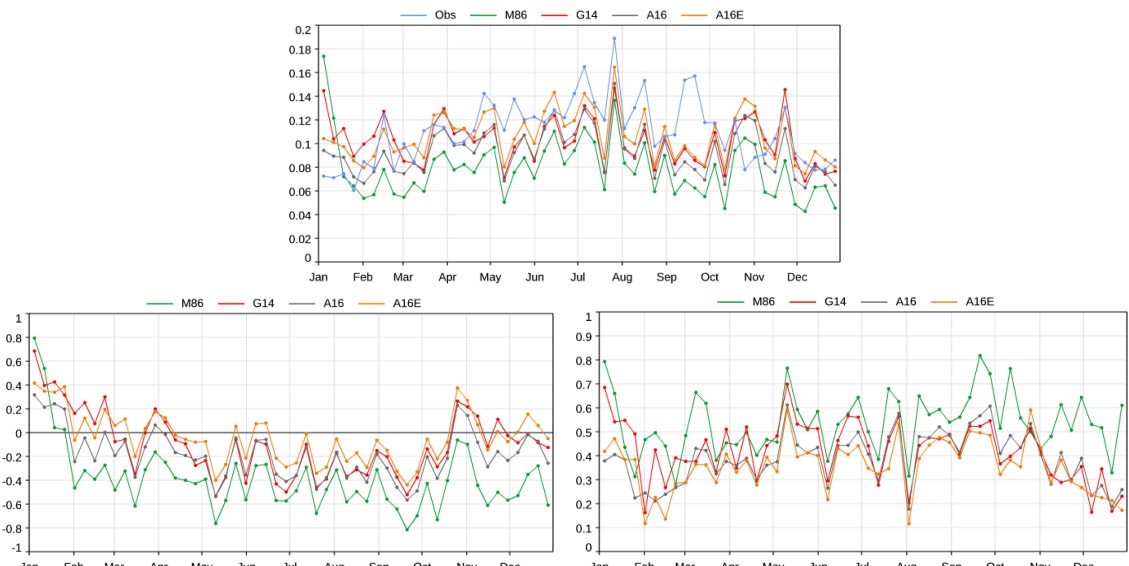

**Figure 9.** Weekly values for observed and simulated total AOD in 2014 over a selection of fourteen AERONET stations representative of sea salt aerosol (top). Bottom left and right: MNMB and FGE, respectively, of weekly simulated total AOD against AERONET AOD averaged for the fourteen stations. Color lines specified in the legends above each panel.

## 4. Discussion

The experiments conducted in this study (Table 1) use different shape functions (size distributions) and scaling factors (magnitudes) in the SSSF (1) employed to obtain sea salt emissions. These changes impact the simulated sea salt surface concentration and AOD (Sections 3.4 and 3.5). We analyze these results to assess the comparative skill of the IFS-AER for each of the tested schemes, and thus identify which scheme improves its performance best. The results can also inform us on the validity and utility of the satellite $W$ retrievals, as well as the relative contributions of size distribution and magnitude to the SSSF. We discuss these three topics here.

### 4.1. IFS-AER Performance with Different Sea Salt Emission Schemes

We start with the results on the AOD. A major reason for emission scheme G14 to be implemented in IFS-AER cycle 45R1 was to remedy the low bias in sea salt AOD simulated by the M86 scheme as compared to both AERONET and MODIS (Moderate Resolution Imaging Spectroradiometer). Figure 7 clearly shows that G14 fulfills this goal. The two new schemes, A16 and A16E, also increase the sea salt AOD above that from M86, but they are still lower than the AOD simulated with G14 (Figure 7). Because all other schemes (but M86) bring the simulated AOD closer to both satellite (Figure 8) and in situ (Figure 9) AOD observations, we now focus on the IFS-AER performance from G14, A16, and A16E.

To further distinguish which schemes lead to best the IFS-AER performance, we need to look at other variables. Looking at SS (Figures 5 and 6), we see that G14 underestimates the observed SS from AEROCE/SEAREX (Figure 5) and overestimates CASTNET SS observations (Figure 6). Meanwhile, A16 and A16E experiments obtain the simulated SS values closer to the 1:1 lines for both data sets. On this basis, here we drop G14 from further consideration and compare and contrast the results from A16 and A16E.

The statistics for AEROCE/SEAREX observations (Figure 5) are best for A16, while for CASTNET (Figure 6) the best is for A16E. To resolve this, we look at Figures 8 and 9 and Table 3. In both cases, it is clear that changing only the scaling factor in experiment A16 yields notable yet less improvement than changing both the scaling and the shape factors in A16E experiment. Therefore, future improvements of the IFS-AER performance in regard to sea salt aerosols should consider improved representations of both the scaling and the shape factors.

From this analysis, we single out sea salt emission scheme A16E as the one yielding the best performance in IFS-AER simulations. On this basis, we recommend emission scheme A16E (Section 2.2.3) for the next update cycle of the IFS-AER.

### 4.2. Validity of Whitecap Fraction from Satellite Observations

In this study, for the first time, we use satellite-based retrievals of whitecap fraction for both simulation and evaluation of sea salt emissions and sea salt aerosol products in a major numerical prediction model. This was the overarching goal behind the effort to develop the whitecap remote sensing technique [18,27]—to produce routine observations of $W$ on a global scale for all seasons, because only such data, representing different conditions and different forcing variables, can help to model and characterize $W$ and its variability.

Validation of the $W$ retrievals lends confidence for their use in numerical models, either directly or via parameterizations of $W$ based on them. The most rigorous validation approach is one-to-one (1:1) comparison of temporally and spatially matched in situ and satellite data [35]. However, 1:1 validation of the $W$ retrievals is a challenge because the available in situ $W$ data are insufficient to represent well characterized ground truth for a wide range of conditions and parameters. Anguelova and Bettenhausen (2019, Section 6 in [27]) point out the reasons why in situ $W$ data do not amount to a data set of reference quality; these are roughly the same as the reasons we could not use in situ $W$ data in this study (Section 2.3.1).

To circumvent this difficulty, ref [27] suggested evaluation of the satellite $W$ retrievals with a variety of reference data. In addition to in situ $W$ data, it is possible to use variables computed with $W$ and compared to time-space matched observations of these variables. We refer to this validation approach as an indirect validation. Sea salt aerosol products (SS and AOD) obtained in this study are examples of variables that can be used for indirect validation of satellite-based $W$ retrievals.

Here, we demonstrate the indirect validation by analyzing how the use of both in situ and satellite $W$ data to simulate SS and AOD affects their comparison to observed SS and AOD. Specifically, we look at whether the satellite $W$ data diminish (or not) the differences between simulated and observed SS and AOD values. If the satellite $W$ data improve the simulated SS and AOD, then we consider the satellite $W$ retrievals indirectly validated.

For this analysis, we use experiments M86 and A16 (Table 1) because they have the same shape function (2) and (3), but different scaling factors implemented with different parameterizations of the whitecap fraction. With these settings, all elements of the IFS-AER simulations are the same but the scaling factor in the SSSF. Therefore, any differences in the IFS-AER results will be due solely to the used $W$ data.

The $W$ values in experiment M86 are obtained with parameterization (4), which is based on in situ photographic $W$ observations. The $W$ values in A16 are from parameterization (7) and (8), which is based on satellite radiometric $W$ retrievals. Both parameterizations compute $W$ values which deviate from what is expected for the dependence of $W$ on $U_{10}$. Physical reasoning shows that $W \propto U_{10}^3$ [36]. However, unless forced, existing $W$ data do not support $W(U_{10})$ parameterizations with cubic dependence of $W$ on $U_{10}$ (e.g., Table 1 in [18], and [37]). The current understanding is that a wind speed exponent different from cubic reflects the natural variability of the whitecap fraction, i.e., the influence of different MetOc factors on the $W(U_{10})$ relationship may enhance it (exponent > 3) or weaken it (exponent < 3).

Parameterization (4) in M86 is strongly non-linear, with wind speed exponent above cubic. This is visualized in Figure 3 (bottom left) with the large latitudinal differences of $W$ values: $W$ is very low under the prevalent low winds at low latitudes and has the highest values under predominantly high wind conditions at high latitudes.

Parameterization (7) and (8) in A16 has a quadratic wind speed exponent, which accounts implicitly for the global average effect of all additional factors (see Section 2.3 in [23]). While various MetOc variables weaken the wind speed dependence below cubic, the specific SST dependence bolster whitecapping in the warm waters with persistent

Trade winds (Figure 1). As a result, we see less latitudinal gradient in $W$ from low to high latitudes (Figure 3, bottom right) in experiment A16 than in experiment M86.

Higher scaling factor in experiment A16 leads to higher sea salt emissions (Table 2). These rectify the underestimation of SS by M86 when compared to AEROCE/SEAREX and CASTNET data (Figures 5 and 6, left panels). Ultimately, the scaling factor of experiment A16 increases the sea salt AOD as compared to AOD predicted with M86 (Figure 7, left panels), bringing it closer to satellite AOD (Figure 8, top and 3rd row panels) and in situ AOD (Figure 9). Therefore, the SSSF changes coming from the use of the $W$ parameterization based on satellite $W$ retrievals improve the simulated SS and AOD compared to those produced with (4). Thus, these results validate the magnitude and the global distribution of $W$ from parameterization (7) and (8).

The observed $W$ values for 2014 (Figure 3 top) deviate from the computed $W$ values in the A16 experiment by, at most, 0.4% (Figure 4, bottom left). Thus, we can expect similar improvement of the sea salt aerosol products from the IFS-AER if $W$ parameterization was developed using the 2014 $W$ retrievals. This indirectly validates the $W$ retrievals from the latest whitecap algorithm (Section 2.3.1). We stress that this is a validation of the magnitude of the $W$ retrievals but not of the models used for those retrievals, i.e., the validity and utility of the models for the permittivity, or sea surface roughness emissivity, or foam emissivity (Section 2.3.1) need to be assessed with different methods and data.

### 4.3. On Scaling and Shape Factors of Sea Spray Generation Function

The experiments in this study (Table 1) were designed to pursue the study goals (Section 1). These experiments are not sufficient to rigorously quantify the relative importance of the shape function and the scaling factor. Still, the results, taken together, can help us surmise how much each component of (1) affects the changes in the IFS-AER aerosol products (SS and AOD).

For this analysis, we use the results from experiments M86, A16, and A16E. The changes from M86 to A16 reflect the influence of using different scaling factors at a fixed shape factor. The changes from A16 to A16E show the influence of the shape functions at fixed scaling factor.

As shown in Section 4.2, the use of (7) and (8) (based on satellite $W$ retrievals) in A16 instead of (4) (based on in situ $W$ observations) in M86 produces a scaling factor higher in magnitude and more uniformly distributed by latitude. Such changes in the A16 scaling factor improve all sea salt aerosol products from the IFS-AER compared to those obtained using the scaling factor in experiment M86.

Experiments A16 and A16E have the same scaling factor (7) and (8) but different shape functions: (2) and (3) in A16 and (9) and (10) in A16E. The shape functions have similar functional forms, but they produce size distributions that differ in both shape and range of modeled sizes of sea salt particles. Experiment A16 does not model the fine particles (bin 1) and models the super coarse particles (bin 3) only partially, while A16E covers well all sea salt bins used in the IFS-AER.

When all relevant sea salt sizes are simulated in experiment A16E, the emissions of fine particles are higher, while the emissions of coarse and super coarse particles are more limited compared to those of the A16 experiment (Table 2). This reflects on the particle lifetimes (Table 2) that, in turn, through transport and deposition, affect sea salt SS (compare Figures 5 and 6, bottom panels) and AOD (Figure 7, bottom panels). Figure 8 (panels on 3rd row versus panels at the bottom) and Figure 9 demonstrate that the use of the expanded size distribution in A16E improves the sea salt AOD relative to that of A16.

Overall, each of the two-step changes—from M86 to A16 to A16E—yields an increase in sea salt emissions which improves the IFS-AER predictions of all aerosol products. Focusing on the FGE metrics (Section 2.3.3) of the total AOD in Figure 8, we see that when changing the scaling factor in the SSSF from (4) to (7) and (8), the FGE changes from a global value of 0.241 in M86 to 0.164 in A16—a 30% improvement. When the shape function is changed, the FGE changes from a global value of 0.164 in A16 to 0.142 for A16E, a 15%

improvement. From these data, the scaling factor seems to be more important than the shape factor.

However, it is possible that altering the order of changes in the experiments—first change the shape function and then the scaling factor—may affect the specific numbers in each step. This is especially true regarding the total AOD, used in the analysis here, because all processes (emission, transport, and deposition) contribute to its values. So, when implemented first, the shape function may bring more improvement than the scaling factor. On this basis, we surmise that both the shape function and the scaling factor are most likely equally important—each decreases the difference between simulated and observed AOD values by roughly the same factor.

## 5. Conclusions

We implemented two new sea salt emission schemes in ECMWF's IFS-AER by introducing a new parameterization of whitecap fraction and a modified sea salt size distribution (Section 2.2.3) in the SSSF (1). We compared the results from the two new schemes to two existing ones (Sections 2.2.1 and 2.2.2) by running the IFS-AER in four experiments: M86, G14, A16, and A16E (Table 1). The simulation results (Table 2, Figures 3 and 7) have been evaluated against in situ and/or satellite observations of whitecap fraction (Figure 4), sea salt aerosol surface concentration (Figures 5 and 6), and total AOD (Figures 8 and 9).

The use of the new parameterization of whitecap fraction clearly improved the skill of all aerosol products as compared to scheme M86. The change of only the SSSF scaling factor in experiment A16 yields some improvement over experiment M86 in terms of simulated sea salt AOD and surface concentration. The whitecap fraction simulated by A16 is also much closer to retrievals: the A16 approach provides a better scaling factor in the context of IFS simulation. Change of both the scaling and the shape factors in experiment A16E resulted in the best IFS-AER performance: the [25] size distribution (9) and (10) gives better results in our configuration than the [20] shape factor (2) and (3). Our analysis shows that the scaling and shape factors could be equally important for the sea salt emissions (Section 4.3). Further improvements of the sea salt contribution to the aerosol products should follow improvements of the scaling or shape factors individually or in tandem.

In addition to the use of a new whitecap parameterization, the use of whitecap retrievals (Section 2.3.1) proves beneficial because it gives the possibility to compare simulated and retrieved whitecap fractions. Such a comparison provides direct feedback to the actual sea salt aerosol emissions, unlike AOD or surface concentration, which are also impacted by other processes such as aerosol deposition and transport.

On the other hand, the comparisons of simulated and observed total AOD provide the possibility for indirect validation of satellite retrievals of whitecap fraction (Section 4.2). This is extremely useful for further development and refinement of the whitecap remote sensing algorithm, as reliable ground truth for whitecap fraction is limited.

**Author Contributions:** S.R. implemented the new parameterization in the IFS, carried out experiments and evaluation, and wrote the manuscript; M.D.A. provided the dataset with whitecap retrievals for evaluation and revised the manuscript. All authors have read and agreed to the published version of the manuscript.

**Funding:** S.R. research is supported by the Copernicus Atmosphere Monitoring Services (CAMS) programme managed by ECMWF on behalf of the European Commission. M.D.A. was funded by the Office of Naval Research, NRL program element 61153N.

**Institutional Review Board Statement:** Not applicable.

**Informed Consent Statement:** Not applicable.

**Data Availability Statement:** Not applicable.

**Conflicts of Interest:** The authors declare no conflict of interest.

## Abbreviations

Here are lists of the acronyms, abbreviations, and variables used in the main text

*Acronyms*

| | |
|---|---|
| AEROCE | Atmosphere/Ocean Chemistry Experiment |
| AERONET | Aerosol Robotic Network |
| AOD | Aerosol Optical Depth |
| CAMS | Copernicus Atmosphere Monitoring Service |
| CASTNET | Clean Air Status and Trends Network |
| CB05 | Carbon Bond Mechanism 5 (a chemistry scheme) |
| CYxxRn (xxRn) | Implementation cycle of ECMWF's IFS |
| ECMWF | European Centre for Medium-Range Weather Forecasts |
| FMI | Finnish Meteorological Institute |
| GDAS | Global Data Assimilation System |
| GEMS | Global and regional Earth-system Monitoring using Satellite and in situ data |
| GHz | GigaHertz |
| IFS | Integrated Forecasting System |
| IFS-AER | Integrated Forecasting System aerosol module |
| IFS-CB05 | Integrated Forecasting System chemistry module |
| LOA | Laboratoire d'Optique Atmosphérique |
| LMD | Laboratoire de Météorologie Dynamique |
| LOA/LMDZ | The general circulation model of LOA and LMD |
| MACC | Monitoring Atmospheric Composition and Climate, series II and III |
| NA | Not applicable (not available) |
| NCEP | U. S. National Center for Environmental Prediction |
| QuikSCAT | Quick Scatterometer |
| SEAREX | Sea/Air Exchange |
| SSSF | Sea Spray Source Function |
| SST | Sea Surface Temperature |

*Abbreviations*

| | |
|---|---|
| A16 | Albert et al. (2016) and sea salt emission scheme based on it |
| A16E | Sea salt emission scheme with Extended size range of applicability |
| G14 | Grythe et al. (2014) and the sea salt emission scheme based on it |
| M86 | Monahan et al. (1986) and the sea salt emission scheme based on it |
| M80 | Monahan and O'Muircheartaigh (1980) |
| MetOc | Meteorological and oceanographic |
| $Cl^-$ | Chlorine ion |
| $Na^+$ | Sodium ion |
| PM10 | Particulate Matter with particle size less than 10 μm |
| RH | Relative Humidity |

*Variables*

| | |
|---|---|
| $A$ | Expression (10) used in (9) |
| $a$, $b$ | Regression coefficients in (7) |
| $a_{0,1,2}$, $b_{0,1}$ | Regression coefficients in (8) |
| $B$ | Expressions (3) and (10) used in (2) and (9), respectively |
| $D_p$ | Dry particle diameter |
| $dF/dr_{80}$ | Sea spray production flux |
| $f_i$ | $i$-th forecast |
| $o_i$ | $i$-th observation |
| $f(p_1, p_2, \ldots)$ | Function of forcing parameters $p_1, p_2, \ldots$ representing the scaling factor (magnitude) in SSSF |
| $g(r_{80})$ | Function of particle radius at RH = 80% representing the shape factor (size distribution) in SSSF |
| FGE | Fractional Gross Error |
| MNMB | Modified Normalized Mean Bias |
| RMSE | Root-mean-square error |

| | |
|---|---|
| $N$ | Number of samples in a population |
| $r$ | Correlation coefficient |
| $r_0$ | Particle radius at formation (RH = 100%) |
| $r_{80}$ | Particle radius at RH of 80% |
| $r_d$ | Dry particle radius (RH = 0%) |
| SS | Surface concentration of sea salt aerosols |
| $T$ | Sea surface temperature |
| $T_B$ | Brightness Temperature |
| $T_W$ | Scaling factor in emission scheme G14 |
| $U_{10}$ | Wind speed at 10 m reference height |
| $W$ | Whitecap fraction |
| $W(U_{10})$ | Wind speed dependence of whitecap fraction |
| $W(T_B)$ | Whitecap fraction retrievals from satellite brightness temperature |
| $\theta$ | Adjustable parameter in expression $A$ |

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
