# Peer review of "Improving the Representation of Whitecap Fraction and Sea Salt Aerosol Emissions in the ECMWF IFS-AER"

_remotesensing, doi:10.3390/rs13234856_

Round 1
Reviewer 1 Report
The study examines four formulations of the white cap fraction and sea salt aerosol emissions in weather forecast model. Good formulations are important for better modelling of air-sea fluxes in ocean- atmosphere models (line 392). Results of model formulations are compared to values from an observation network (the CASINET network). However, the observed ground truth values are limited in their reliability (line 508). The authors find that the A16E construct give the best result (line 378), but that it could be improved by better formulations of two partial constructs, a scaling factor and a shape function. The scaling factors, x, is part of the wind speed construct, U10x and the shape function, dF/dDp, (representing the sea spray size distribution) is the derivative of F with respect to the dry particle parameter, Dp. To find the best formulation they use four statistical measures, among them RMSE, and OLR regression coefficients (Figure 5).
The paper is rather technical with many references that have to be consulted to be able to duplicate the study. However, the paper is also rather long, and thus references might do the job. The paper is well written, easy to follow (apart from all the required details). Since the paper is already long, I would hesitate to suggest more figures, but I did think that it would be nice with graphs showing the effects of the different scaling factors. Maybe it could be depicted in contour graphs with U and Dp on the axes, I am not sure it will work.
The authors recommend the A16 E formulation, but maybe they could add the values of the “best” -their assessment- of the scaling factor and the shape factor. I guess I could find out, but to me it is not obvious.
Not for the authors to do, but
- it would have been nice with some error measures for all the parameters
- Would all the parameters really be significant?
- I started counting all the parameters for the four methods, but gave in. However, with parsimonious in mind, would it be helpful to have a count of the parameters, and comment on this in the discussionsection: The method A16E is recommended, and it is also the method that depends on least number of parameters?
- In hydrodynamics, a method depends crucially on one numerical construct. The method is applied to hydrodynamic problems all over the word. However, the parameters for the construct was determine by some observations taken from deserts in Arizona. So, what type of data sets have been used to parametrize the equations? And from which region?
- It is difficult not to wonder if all these methods could be simplified, but again, this would be outside the scope of the present study.
Minor comments
It is so many parameters here. I think it would be advantageous with a list of acronyms – if the journal allows it
Line 77. RH = 0%, means what, I suppose relative humidity, but should be spelled out
Line 78 by a factor of 4.3 →is this obvious?
Line 109 “hereafter” you say this also on line 2018
Line 173-4. It is not quite clear to me when satellite and when in-situ observations are used.
Eqs. (12) and (13) are these formulations really required? Figures 4 and 5 uses ordinary RMSE and OLR
Line 266. Eqs. 7 and 8?
Figure 2 Units for the color bar
Line 316 What is the range of simulated (modeled??) and retrieved (observed??) values; 0 to 50 and 0 to 500 respectively?
Figures 4 and 5 Nice Figures, Should you add probability values, p.
Figure 6 Horizontal bar below figure
Line 364. Closer to both →closer to observations from both.. . It is good to stress – with the same terms – if you talk about modeled / simulated or observed data
Line 378. Isn’t this discussion matter.
Line 366. You do not need “In this study...”
Line 391. This was a nice formulation of the objective. Maybe move it forward
Line 422. Different from cubic... Here I was inspired to ask for a comparison between “natural” parameter values and the ones in the equations.
Line 428. “ .. which accounts implicitly for the global average effect of all additional...” This was a little unsatisfactory for me, but again, it may be outside the scope of this study.
Author Response
Dear reviewer,
Thanks for your careful review. Please find attached our answers to your remarks and comments.
Kind regards,
Samuel Rémy
Reviewer 2 Report
Please see attached file.

Author Response
Dear reviewer,
Thank you for your careful review. Please find attached our answers to your comments and remarks.
Kind regards,
Samuel Rémy

Round 2
Reviewer 2 Report
This reviewer is now satisfied with the revisions the authors have now made to this manuscript.